# Pharmacogenetic Approach to Tramadol Use in the Arab Population

**DOI:** 10.3390/ijms25168939

**Published:** 2024-08-16

**Authors:** Chan-Hyuk Kwon, Min Woo Ha

**Affiliations:** 1Seoul Shingil Rehabilitation Medicine Clinic, 162 Shingil-ro, Yeongdeungpo-gu, Seoul 07362, Republic of Korea; 2Jeju Research Institute of Pharmaceutical Sciences, College of Pharmacy, Jeju National University, 102 Jejudaehak-ro, Jeju-si 63243, Jeju-do, Republic of Korea; 3Interdisciplinary Graduate Program in Advanced Convergence Technology & Science, Jeju National University, 102 Jejudaehak-ro, Jeju-si 63243, Jeju-do, Republic of Korea

**Keywords:** tramadol, pharmacogenetics, CYP2D6, OPRM1 A118G, Arab

## Abstract

Tramdol is one of most popular opioids used for postoperative analgesia worldwide. Among Arabic countries, there are reports that its dosage is not appropriate due to cultural background. To provide theoretical background of the proper usage of tramadol, this study analyzed the association between several genetic polymorphisms (CYP2D6/OPRM1) and the effect of tramadol. A total of 39 patients who took tramadol for postoperative analgesia were recruited, samples were obtained, and their DNA was extracted for polymerase chain reaction products analysis followed by allelic variations of CYP2D6 and OPRM A118G determination. Numerical pain scales were measured before and 1 h after taking tramadol. The effect of tramadol was defined by the difference between these scales. We concluded that CYP2D6 and OPRM1 A118G single nucleotide polymorphisms may serve as crucial determinants in predicting tramadol efficacy and susceptibility to post-surgical pain. Further validation of personalized prescription practices based on these genetic polymorphisms could provide valuable insights for the development of clinical guidelines tailored to post-surgical tramadol use in the Arabic population.

## 1. Introduction

In contemporary society, pain affects more individuals than any other medical condition, leading to significant economic losses [1]. Consequently, understanding the pathophysiology of pain and effective pain management has become increasingly important. Recent advancements in genetics have been integrated into pain studies, with efforts focusing on personalized medicine through the exploration of pain-related genes [2,3,4,5]. Tramadol is one of the most commonly used opioids for postoperative analgesia [6,7]. Despite its widespread use globally, reports indicate that in Arab countries, its administration at appropriate dosages is challenging due to religious and cultural factors. This issue, coupled with the recent advancements in medical technology, has highlighted the need for scientific discussions on the proper use of opioids in the affluent Middle Eastern countries, where there is an increased interest in enhancing the quality of life [8,9].

Tramadol is metabolized into its active form, O-desmethyltramadol, by the enzyme CYP2D6. Thus, different metabolic phenotypes, which are determined by CYP2D6 genetic polymorphisms, influence its pharmacologic effects [10,11,12]. Additionally, certain CYP2D6 alleles associated with distinct metabolic phenotypes exhibit notable ethnic variations, although studies focusing on the Arab population are relatively scarce compared to others [13,14,15].

Tramadol exerts its analgesic effects by binding to opioid receptors, with the μ-opioid receptor (MOR), encoded by the OPRM1 gene, playing a key role in this process. The OPRM1 gene, located on chromosome 6, has more than 3000 identified polymorphisms. Among these, rs1799971 (OPRM1 A118G) is the most extensively studied single nucleotide polymorphism. Studies have shown that patients with the G allele may require higher doses of tramadol to achieve adequate analgesic effects [7,16,17]. To provide a scientific foundation for the appropriate use of tramadol, this study explored the relationship between genetic phenotypes (CYP2D6 and OPRM1 A118G) and the effect of tramadol in Arabs residing in the United Arab Emirates.

## 2. Results

### 2.1. CYP2D6 Allele Frequency

Among the 39 volunteers, 10 different CYP2D6 alleles were identified. The CYP2D6*1 allele had the highest frequency, occurring in 43% (*n* = 34) of the subjects. This was followed by CYP2D6*2 and CYP2D6*41, with frequencies of 18% and 10%, respectively (Table 1).

In this study, the frequency of functional alleles was 65.4%, while reduced-function and non-functional alleles were observed at frequencies of 21.8% and 12.8%, respectively. The prevalence of duplicated CYP2D6 genes was 5.12%, with the specific frequencies being 2.56% for *1xN, 1.28% for *2xN, and 1.28% for *4x2 (Figure 1).

### 2.2. CYP2D6 Metabolic Phenotype and Tramadol Effect

Among the 39 volunteers, the CYP2D6 metabolic phenotypes were classified as intermediate metabolizers (IM), normal metabolizers (NM), and ultrarapid metabolizers (UM), with 11 (28.2%), 25 (64.1%), and 3 (7.7%) subjects, respectively. No poor metabolizers (PM) were observed in this study.

The effects of tramadol, measured as the difference in NRS scores (ΔNRS), were 2.45 for IM, 3.28 for NM, and 4.67 for UM. These values were significantly different according to the metabolic phenotype (Figure 2).

### 2.3. OPRM1 A118G Allele Frequency and the Analgesic Effect of Tramadol in Relation to OPRM1 Genotype

Among the 39 Arabian subjects tested, 25 subjects were identified to have the homozygous OPRM1 A118A genotype (64.1% of the tested subjects), and 14 subjects were heterozygous for the OPRM1 A118G genotype (35.9%) in this study. Therefore, the minor allele frequency, the “G allele”, was calculated to be 17.95% (Table 2). No homozygous OPRM1 G/G genotype was found in this study.

The basal pain score showed a statistically significant difference between subject groups, with the details of OPRM1 G118G for 9.14 and OPRM1 G118A for 8.04 (*p* < 0.01) in this study (Figure 3). However, such a difference was not observed for the analgesic effect of tramadol (ΔNRS) between these genotype groups in this study (Table 2).

### 2.4. Adverse Drug Reactions of Tramadol

In this study, no patient reported any adverse drug reactions to tramadol following the administration of a single dose of 100 mg tramadol. All 39 patients received only one dose of tramadol, and none requested a second dose or an alternative painkiller. There were no reported side effects observed in this study.

## 3. Discussion

Pain affects more than 100 million people in the United States alone, surpassing the combined numbers of patients with cardiovascular disease, cancer, and diabetes, leading to significant economic burdens [1]. In response, efforts in the field of pharmacogenetics have been directed towards identifying candidate genes for pain and providing personalized medicine. Pain represents a quintessential case of genetic–environmental interaction and is a focal point of interest in genetic medicine [2]. Recent advancements in genetic studies have identified at least 358 genes considered to be related to pain [18], leading to an explosive increase in genetic relationship research and efforts to pinpoint pain-related genes [19,20].

Tramadol stands as one of the most widely used opioids for postoperative pain control due to its relatively lower side effects, such as respiratory suppression. Acting on the central nervous system (CNS) as a μ-receptor specific weak agonist and a norepinephrine/serotonin (5-hydroxytryptamine; 5-HT) reuptake inhibitor [21], tramadol’s efficacy is noted. However, in Muslim countries where there is a taboo on substances altering mental states, using the appropriate dose of opioids such as tramadol has proven difficult, necessitating scientific discourse [8,9]. In these nations, opioids like tramadol are stringently regulated by federal law, limiting access to such medications. This limitation has led to illicit means of obtaining medications and reluctance among patients to use controlled substances. Moreover, in very conservative Muslim communities, pain may be viewed as a form of punishment for sin, leading to reluctance to request analgesics or increase their dosage despite inadequate pain control. These attitudes are particularly pronounced among poorer and less educated individuals, leading to inadequate pain treatment [8].

CYP metabolizes approximately 20–25% of all prescribed medicines [22] and plays a crucial role in about 80% of total phase I drug metabolism [23]. CYP2D6, located in a gene cluster with frequent variations, has more than 100 reported variants [24]. It is the sole active CYP2D gene in humans [25]. CYP2D6 alleles are associated with four metabolic phenotypes, with *1 and *2 being functional alleles, while *4 and *5 are non-functional alleles, and *10, *17, and *41 are considered reduced-function alleles. Some alleles exhibit distinct racial frequency differences. For instance, CYP2D6 *4, an allele lacking CYP2D6 function, is associated with slow metabolism and is found at a frequency of about 15% in Caucasians [26], but only about 1% in East Asians [27]. Studies of CYP2D6 alleles in the Arab population exist [28], but are insufficient for conclusive findings. Previous studies reported average CYP2D6 allele frequencies across different ethnic groups (Table 3) [29,30]. Although specific numbers vary, similar trends were observed between previous Arabic CYP2D6 allele frequencies and those found in this study. In the Middle East region, CYP2D6 *1 was the most frequent allele, followed by *2 and *41. In this study, *4 was found with a frequency of 10.26%, lower than in Caucasians but higher than in other Asian populations. The frequency of non-functional alleles was 12.8%, again lower than in Caucasians but higher than in other Asian populations. The frequency of copy number variants for deletions and multiplications in the CYP2D6 gene is an important factor in determining metabolic phenotypes. Patients with multiple copies of functional CYP2D6 alleles may exhibit higher rates of drug metabolism. In this study, the number of duplicated alleles was slightly higher than in other Asian populations.

Tramadol serves as a substrate for assessing CYP2D6 activity, undergoing metabolic conversion to O-desmethyltramadol, which exhibits pharmacological activity. Thus, metabolic phenotypes determined by CYP2D6 genotype impact the pharmacological efficacy of tramadol [10,11,12]. Individuals with the CYP2D6 poor metabolizer (PM) genotype exhibit lower analgesic effects at the same dose compared to those with the extensive metabolizer (EM) genotype [31], while patients with the ultrarapid metabolizer (UM) genotype may experience improved analgesic effects but a higher incidence of side effects [32]. In this study, the relationship between tramadol efficacy and CYP2D6 metabolic phenotype mirrored previous findings. However, assessing the relationship between side effects and CYP2D6 was challenging due to the absence of reported side effects. This is likely because all patients received only one dose of tramadol, and the dose required to induce side effects was not administered. Additionally, in this study, no additional invasive tests could be performed beyond those required for patient treatment. Consequently, we were unable to measure the plasma concentrations of tramadol and its active metabolite through blood sampling, which posed a limitation in interpreting the drug’s efficacy and side effects in this research. For this reason, it is challenging to determine whether all patients requested only a single tramadol injection because it achieved sufficient plasma concentrations of tramadol and its active metabolite, or if the patients were merely enduring the pain. Nonetheless, the relatively high mean pain score of 5.28 ± 2.05 cautiously suggests the possibility that adequate pain management may not have been achieved.

The opioid receptor, a member of the Rhodopsin family of G-protein coupled receptors, initiates downstream signaling. The μ-opioid receptor (MOR), encoded by the OPRM1 gene, is recognized for its role in analgesic response [33]. Painful stimuli trigger the release of endogenous opioids, activating MOR to produce analgesic effects. The most studied OPRM1 single nucleotide polymorphism (SNP) is rs1799971, known as A118G, which is associated with several functional differences. The G allele of A118G introduces a novel CpG methylation site, inhibiting upregulation of OPRM1 during chronic opioid use [34]. Studies have shown reduced mRNA copy numbers carrying the G allele in human brain tissue compared to the A allele [35], and reduced cell surface expression of MOR with the A118G variant in stably transfected cell lines [36]. Consequently, investigations into the relationship between OPRM1 polymorphism and pain sensitivity have been conducted. Carriers of the G allele have been reported to exhibit diminished opioid analgesic effects in numerous studies [37,38,39,40]. However, despite substantial evidence supporting the role of A118G in opioid analgesics, some studies have failed to find this association [41,42,43,44]. Moreover, studies have demonstrated that women with the A/A genotype require higher opioid doses to achieve analgesic effects during labor [45,46]. These conflicting findings underscore the complexity of pain control. Meanwhile, efforts have been made to examine OPRM1 polymorphism and pain threshold. Women with the G allele have reported higher pain scores in migraine pain [47], experienced more pain in disk herniation [48], and suffered more from pain in fibromyalgia [49]. A similar relationship between OPRM1 polymorphism and pain threshold was observed in this study. However, there are also counterexamples to the study of pain threshold [50], and due to the limited number of participants, the homozygous OPRM1 G/G genotype was not observed in this study. Additionally, since the study was conducted at a single hospital, it does not represent all Arabic subpopulations, and all patients received only a single dose of tramadol, which limits the ability to draw clinical conclusions regarding the effects of tramadol based on genotype. Therefore, to derive clinically meaningful conclusions that can be applied in precision medicine, future research should aim to secure an adequate sample size encompassing diverse Arabic subpopulations and conduct an integrated analysis of a broader range of genes. It is also advisable to refer to the recently developed international reporting guidelines [51] from the research planning stage, as this would contribute to synthesizing disparate studies and deriving high-quality conclusions.

## 4. Materials and Methods

### 4.1. Study Subjects

Individuals who received tramadol for postoperative analgesia were recruited from H.H. Sheikh Khalifa Specialty Hospital (SKSH) in the UAE. Those who were unable to communicate properly due to cognitive dysfunction or who declined to participate in the study were excluded.

Following approval from the institutional review boards of H.H. SKSH, a total of 47 patients were enrolled, all of whom provided informed consent. Among these, 8 patients withdrew their consent, resulting in a final sample size of 39 patients. Of these 39 patients, 10 had undergone orthopedic surgery and 29 had undergone general surgery. The mean age was 48.90 ± 15.78 years, and the mean body weight was 82.85 ± 16.87 kg. The study cohort comprised 18 males and 21 females (Table 4).

Basal pain score was measured one day after the removal of patient-controlled analgesia using a numerical rating scale (NRS: 0–10, where 0 indicates no pain and 10 indicates the worst pain imaginable). This measurement was taken before administering any analgesics. The pain score was measured again one hour after administering 100 mg of tramadol intravenously. The effect of tramadol was assessed by calculating the difference in NRS scores (ΔNRS) between the baseline and one hour post-administration.

All study subjects were also retrospectively evaluated for the presence of adverse drug reactions to tramadol using the electronic medical record system. Additionally, medication prescription history for side effects was carefully reviewed.

### 4.2. Genotyping of CYP2D6 and OPRM1

Genomic DNA was extracted from dried blood spots (DBS) (Figure 4) using the FTA Purification Reagent (GE Healthcare, Chicago, IL, USA). These samples were collected during routine laboratory tests following surgery.

Primers for CYP2D6 genotyping were prepared according to the method published by Kim et al. [13]. Polymerase chain reaction (PCR) was performed using the GeneAmp PCR9700 (Applied Biosystems, Foster City, CA, USA) with a 30 µL mixture containing 100 ng genomic DNA, 10× EF-Taq Reaction Buffer (25 mM MgCl_2_), 2.5 mM dNTPs, primers, 5× Band Doctor (SolGent Inc., Daejeon, Republic of Korea), and EF-Taq DNA Polymerase (2.5 U/µL). Predenaturation, denaturation, annealing, and extension were conducted at 94 °C for 5 min, 98 °C for 20 s, 64 °C for 30 s, and 72 °C for 5 min, respectively, and were repeated for 35 cycles. The final extension was performed at 72 °C for 5 min.

Primers for OPRM1 genotyping were prepared following Kim et al. [52]. The PCR reactions, including predenaturation, denaturation, annealing, and extension, were conducted at 95 °C for 2 min, 95 °C for 20 s, 64 °C for 30 s, and 72 °C for 30 s, respectively, and were repeated for 35 cycles. The final extension was performed at 72 °C for 5 min.

After obtaining the PCR products, a modified SNaPshot^®^ protocol (Applied Biosystems, Foster City, CA, USA) was applied. PCR products were treated with Exo-SAP IT™ at 37 °C for 35 min and 80 °C for 15 min as a clean-up step. SNaPshot reactions were performed using the SNaPshot™ multiplex kit (Applied Biosystems, Foster City, CA, USA) under the conditions of 96 °C for 10 s, 50 °C for 5 s, and 60 °C for 30 s for 40 cycles. Shrimp Alkaline Phosphatase treatment (37 °C for 60 min, 60 °C for 15 min) was applied as a PCR clean-up method. The final products were analyzed using the 3500 DX Series Genetic Analyzer (Applied Biosystems, Foster City, CA, USA) after heat denaturing with a mixture of Hi-Di formamide and ABI Gene Scan 120Liz at 95 °C. Genotypes of CYP2D6 (according to the CYP2D6 Nomenclature Database) and OPRM1 A118G were identified using GeneMapper software (6.0, Applied Biosystems, Foster City, CA, USA).

### 4.3. Data and Statistical Analysis

The analgesic effect of tramadol was estimated by ΔNRS, which is the difference in NRS pain scores between the baseline measurement taken before tramadol injection and the measurement taken one hour after 100 mg of tramadol IV injection. All NRS scores and other clinical measurements were described using descriptive statistics, such as mean ± SD. The frequency of genotypes and alleles was also estimated among the study subjects.

In this study, we determined the CYP2D6 phenotypes based on the activity scores of their respective genotypes using the method developed by Gaedigk et al. [53]. This method employs a modified Delphi approach to achieve consensus among an international panel of CYP2D6 experts on a standardized system for translating CYP2D6 genotypes into phenotypes. Metabolic phenotypes were classified based on an “activity score”, with each allele assigned an “activity value” ranging from 0 to 1 (e.g., 0 for no function, 0.5 for decreased function, and 1.0 for normal function). The “activity score” was calculated by summing these activity values. Poor metabolizers (PM) were defined by an activity score of 0. Intermediate metabolizers (IM) had activity scores greater than 0 but less than 1.25. Normal metabolizers (NM) had activity scores ranging from 1.25 to 2.25. Ultrarapid metabolizers (UM) were defined by activity scores greater than 2.25. Statistical analysis was performed using SPSS version 18.0 (Chicago, IL, USA). The effect of CYP2D6 metabolic phenotype on the analgesic effect of tramadol was analyzed using the Kruskal–Wallis test and the Mann–Whitney test with Bonferroni adjustment for post hoc analysis. The differences in tramadol effect according to CYP2D6 metabolic phenotypes and OPRM1 A118G alleles were assessed using the Mann–Whitney test.

## 5. Conclusions

This study represents an exploratory endeavor, revealing a significant correlation between the metabolic phenotype of CYP2D6 and the efficacy of tramadol. Additionally, carriers of the G allele of OPRM1 A118G were found to be more susceptible to postoperative pain. Consequently, these genetic polymorphisms emerge as crucial factors in personalized medicine aimed at optimizing the efficacy of tramadol for postoperative analgesia.

In the future, further clinical trials are warranted to validate personalized prescription practices. This exploratory study may serve as a catalyst for the development of clinical guidelines tailored to post-surgical tramadol usage within the Arabic population.

## Figures and Tables

**Figure 1 ijms-25-08939-f001:**
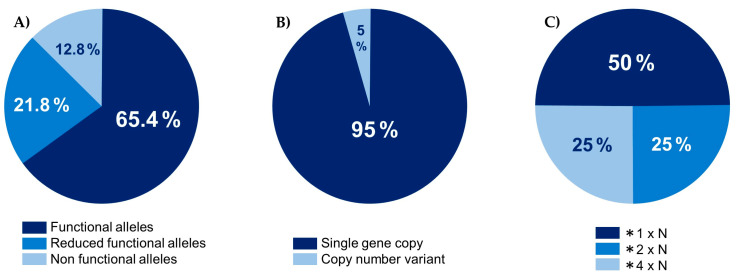
(**A**) Frequency of functional, reduced functional, and non-functional alleles; (**B**) frequency of alleles with a single gene copy (*n* = 74) and those with copy number variants (*n* = 4); (**C**) details of copy number variants.

**Figure 2 ijms-25-08939-f002:**
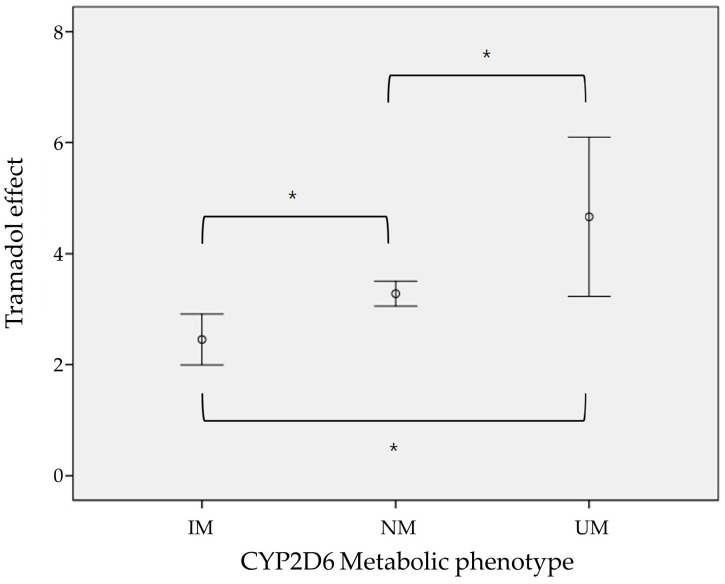
Difference of tramadol effect (ΔNRS) according to metabolic phenotype (*: *p* < 0.01).

**Figure 3 ijms-25-08939-f003:**
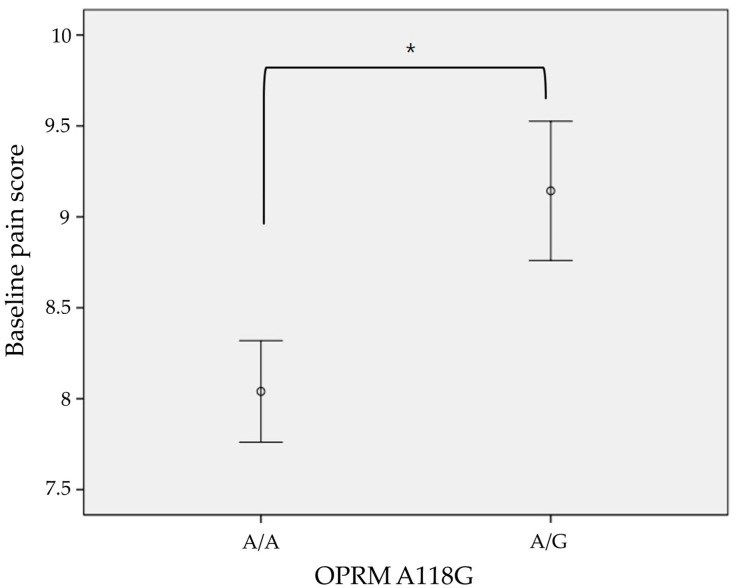
Baseline pain score according to OPRM A118G genotype (*: *p*< 0.01).

**Figure 4 ijms-25-08939-f004:**
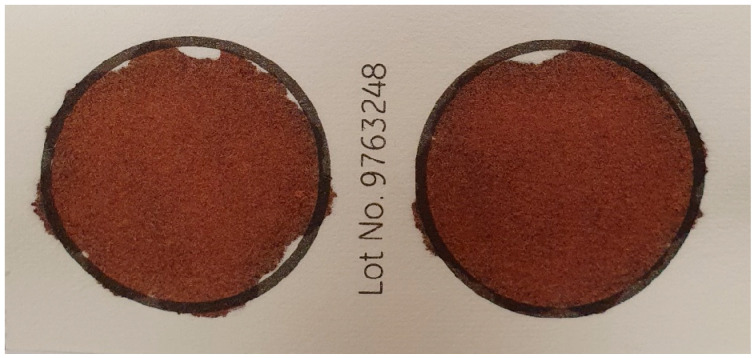
Dried blood spot samples used for the measurement of CYP2D6 and OPRM1 genotypes in this study.

**Table 1 ijms-25-08939-t001:** CYP2D6 allele frequencies.

Alleles	Allele Frequency (%)
CYP2D6*1	34 (43.59%)
CYP2D6*2	14 (17.95%)
CYP2D6*4	8 (10.26%)
CYP2D6*5	1 (1.28%)
CYP2D6*10B	5 (6.41%)
CYP2D6*17	4 (5.12%)
CYP2D6*41	8 (10.26%)
CYP2D6*1xN	2 (2.56%)
CYP2D6*2xN	1 (1.28%)
CYP2D6*4x2	1 (1.28%)
Total	78 (100%)

**Table 2 ijms-25-08939-t002:** Frequency of OPRM A118G genotype and the effect of tramadol (ΔNRS) in relation to OPRM genotypes.

	A/A (%)	A/G (%)	G/G (%)	MAF (%) ^1^
Frequency	25 (64.10)	14 (35.90)	0 (0)	17.95
Tramadol effect (ΔNRS)	3.16	3.14	NA ^2^	

^1^ MAF: minor allele frequency, ^2^ NA: not available.

**Table 3 ijms-25-08939-t003:** Frequency of CYP2D6 alleles in different ethnic groups.

CYP2D6 Alleles	Allele Frequencies (%)					
	East Asian	South/Central Asian	Middle East	Oceania	Caucasian	African
*1	34.2	53.7	58	70.2	39	32.8
*2	12.7	31.9	21.7	1.2	28.1	20.1
*3	0	0.03	0.08	0	1.33	0.03
*4	1.1	6.6	7.8	1.1	18.0	3.4
*5	5.6	2.5	2.3	5	2.8	6.1
*10	42.3	19.8	3.5	1.6	2.9	6.8
*14	0.9	0	0	0	0	0.3
*17	0.01	0.2	1.6	0.05	2.9	20.0
*29	0	0.1	0.8	0	0.1	10.3
*35	0.2	NA ^1^	2.0	0	5.8	NA
*36	1.6	NA	0	0	0	0
*41	2	10.5	20.4	0	7.7	10.9
*XN	0.4	0.5	3.9	0	2.7	7.6

^1^ NA: not available.

**Table 4 ijms-25-08939-t004:** Patient characteristics and clinical features.

Variable		
Age (year)		48.90 ± 15.78
Sex (%)	Male	18 (46.15)
	Female	21 (53.85)
Weight (kg)		82.85 ± 16.87
Height (m)		161.10 ± 8.56
BMI (kg/m^2^)		31.98 ± 6.66
Type of surgery (%)	Orthopedic surgery	10 (25.6)
	General surgery	29 (74.4)
Smoking history		64.10

## Data Availability

The data presented in this study are available on request from the corresponding author due to privacy.

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
