# Peer review of "Pharmacogenetic Approach to Tramadol Use in the Arab Population"

_ijms, 2024, doi:10.3390/ijms25168939_

Round 1
Reviewer 1 Report
Comments and Suggestions for Authors
During drug therapies, it is important to know and understand the role of the genetic variations present in the patients, the environmental background, and even the cultural and religious factors affecting the community, which together can influence the effectiveness of the proper treatments. Nowadays, it has come to the attention of scientific interest to understand the relationships and interactions between genetic variations, environmental triggers and even cultural and religious factors which involved in diseases. Recently, the so called genome-wide association studies (GWAS) enable genetic features to be linked to many disease conditions and the treatment choices. Analyzing the genomic background and the so-called metabolic phenotype of individuals or populations, information can be applied to personalized medicine or public healthcare. The opioid tramadol is among the most frequently used analgesic compound. One important area of ​​its use is post-operative pain relief. Some previous reports indicated that in Arab countries, the administration of tramadol was challenging due to different reasons including the genetic background of the population. In the submitted work of Kwon and Ha a significant correlation between the metabolic phenotype of CYP2D6 (a gene encoding the enzyme involved in tramadol metabolism) and the efficacy of tramadol has been described. Additionally, carriers of the G allele of OPRM1 A118G were shown to be at least partly liable to post-operative pain. The observed genetic polymorphisms emerge as crucial factors in personalized medicine aimed at optimizing the efficacy of tramadol in the treatment of post-operative pain. Although the results presented in the article do not seem particularly significant, moreover, the analysis was based on a relatively small number of patients, I support the publication of these results and do not consider any changes in form or content necessary.
Author Response
Response:
Thank you for your detailed review and the valuable insights provided on our manuscript. We are grateful for the time you dedicated to our study and appreciate your supportive stance regarding publication.
We acknowledge the point raised about the perceived marginal significance of the results and the limitations posed by the size of our patient cohort. While our findings may not be universally definitive, we believe they add important preliminary data to the existing body of knowledge and emphasize the need for further research in this area.
Thank you once again for your constructive feedback and endorsement of our work. We eagerly anticipate the publication of our manuscript in IJMS and believe that it will foster meaningful discussions within the scientific community.

Reviewer 2 Report
Comments and Suggestions for Authors
Dear Authors,
the topic of the manuscript is of interest, however, there are several aspects requiring revisions:
- Please clarify how you define IM, NM, or UM. The % seems different from figure 2.
- The peculiarity of this retrospective study (limitation) is that patients took only 1 dose of tramadol, and none asked for additional painkillers, this seems really strange for post-surgery patients. Can the author double-check if the patients took any other medications? If no other medications were taken, it means that for these patients, no matter what is their genetic background a single dose of tramadol was sufficient. In other words, this diminishes the clinical relevance of this study, that preserves only an interest from a mechanistic point of view.
- In this context, however, the main problem with CYP2D6 is that there is no clear linear relationship between genotyped-determined functions and NRS response to tramadol. Indeed intermediate metabolizers should be between Normal and ultra metabolizers. Is that so?
- The authors should also try to analyze the results of tramadol response combining variation in the two loci (e.g. with a genetic risk score).
- The discussion should be more focused on the results of this study, highlighting strengths and weaknesses.
Author Response
We deeply appreciate your request for revisions and the accompanying comments. We have endeavored to revise the manuscript meticulously in alignment with your suggestions. The amendments have been made accordingly and are highlighted in the revised manuscript for your review. We provide specific details for each item as follows:
# Comments 1; Please clarify how you define IM, NM, or UM. The % seems different from figure 2.
â–¶ Response; Thank you for your insightful comments on our study. IM, NM, UM are phenotypes for CYP2D6 set according to the method by Gaedigk A et al. in reference [19]. As you pointed out, to aid the understanding of general readers, we have added a brief description of this method in the relevant section of the text. Figure 2 shows the allele frequency of the CYP2D6 genotype, indicating which allele is dominant within the gene pool. As per your suggestion, we have included the frequency of each in table 2 for readers interested in the details. Additionally, to prevent confusion between phenotype and genotype among general readers, we have directly described the frequency of the phenotype in the section "B. CYP2D6 metabolic phenotype and tramadol effect."
# Comments 2. The peculiarity of this retrospective study (limitation) is that patients took only 1 dose of tramadol, and none asked for additional painkillers, this seems really strange for post-surgery patients. Can the author double-check if the patients took any other medications? If no other medications were taken, it means that for these patients, no matter what is their genetic background a single dose of tramadol was sufficient. In other words, this diminishes the clinical relevance of this study, that preserves only an interest from a mechanistic point of view.
â–¶ Response; Thank you for your valuable comments. The concerns you have raised were indeed shared by the researchers and served as a motivation for this study. Clinically, it is evident that a variety of medications are necessary for post-operative pain management, yet it has been observed that patients did not request additional medication beyond what was administratively provided by their physicians. Notably, even with considerable pain as indicated by the pain scores recorded after medication input, many patients chose to endure the discomfort without further requests for relief (this observation was substantiated through nursing records and, as it was not quantified, it was not mentioned explicitly in the text). We hypothesize that this behavior might be influenced by the hospital’s operational context, predominantly managed by foreigners with different cultural and religious backgrounds, which could lead to a gap in understanding between patients and physicians regarding treatment options. This discussion is intended to potentially reduce oversight by doctors and alleviate patients' fears regarding treatment protocols. It is pertinent to note that in the countries where the hospital's physicians were trained, tramadol is commonly explained as a "narcotic" painkiller, which might contribute to the reservations about additional medication usage among devout patients in the region. The risk of selection bias due to recruiting participants from only one hospital was acknowledged and documented in L-274.
# Comments 3. In this context, however, the main problem with CYP2D6 is that there is no clear linear relationship between genotyped-determined functions and NRS response to tramadol. Indeed intermediate metabolizers should be between Normal and ultra metabolizers. Is that so?
â–¶ Response; Thank you for dedicating time and thought to this research. In the case of CYP2D6 phenotypes, namely IM, NM, and UM, significant differences in the effect of tramadol have been observed. These findings have been documented in the section "B. CYP2D6 metabolic phenotype and tramadol effect" and illustrated in Figure 3.
Regarding the candidate gene for pain sensitivity, the OPRM1 A118G allele, there were no observed differences in tramadol effect based on genotype. This aligns with findings from references 39-52, suggesting that more research is needed before definitive conclusions can be drawn about this allele. However, as shown in Figure 4 of this study, individuals with the G allele exhibited higher baseline pain scores, which will provide important insights for future research exploring the relationship between the OPRM1 A118G allele and pain sensitivity.
#4. The authors should also try to analyze the results of tramadol response combining variation in the two loci (e.g. with a genetic risk score).
â–¶ Response; Thank you for your astute observations. Calculating the genetic risk score presented considerable challenges within the setting of this study. Additionally, due to the many diverse genotypes of CYP2D6, direct analysis of specific alleles was difficult. Hence, as is common in this field, we analyzed using the phenotypes of CYP2D6 and documented each allele's frequency. Regarding the analysis of various candidate genes for pain sensitivity, our study measured only the OPRM1 A118G, which regrettably limits the scope of our genetic evaluation. Looking forward, if ample subjects are secured and multiple candidate alleles are measured, and if the effect sizes of risk alleles related to pain are established through GWAS, as you mentioned, subsequent research could indeed contribute to the advancement of precision medicine. We have noted these limitations in L-272.
#5. The discussion should be more focused on the results of this study, highlighting strengths and weaknesses.
â–¶ Response; Thank you for your incisive comments. The researchers in this study hoped to engage as many physicians working in the Arab region as possible with this issue. Therefore, we aimed to include the most basic information so that even those without genetic knowledge could understand if they took the time to read it, which may have made our explanation appear somewhat verbose to experts like yourself. This study primarily addresses the phenotype of CYP2D6 and rs1799971 (A118G), the most frequently researched SNP in OPRM1. To reduce confusion among our readers, we have separately discussed the interpretations of these two areas of study in the Discussion section. Additionally, as you pointed out in item 4, we have acknowledged one of the primary limitations of this study—the small sample size—as documented in L-272.
Thank you for your time and evaluation of our manuscript. We have done our best to reflect on your points thoroughly. We wish peace and wellness to your family.
Warm regards,
Min Woo Ha

Reviewer 3 Report
Comments and Suggestions for Authors
Author Response
We deeply appreciate your request for revisions and the accompanying comments. We have endeavored to revise the manuscript meticulously in alignment with your suggestions. The amendments have been made accordingly and are highlighted in the revised manuscript for your review. We provide specific details for each item as follows:
# Comments 1~3; The study only included 39 patients, which limit the generalizability of the findings. This must be discussed as a limitation of the study; The research was conducted at a single hospital, which may introduce locationspecific biases and limit the applicability of the result to other settings. Including multiple hospitals across different regions would help mitigate such biases and enhance the representativeness of the finding. This must be discussed; Similarly, incorporating a wider range of genetic polymorphisms and including diverse subpopulation within the Arab demographic could uncover additional factors influencing tramadol metabolism and response. This must also be discussed;
â–¶ Response; Thank you for your valuable remarks. We have added the mentioned details to L-272.
# Comments 4; Plasma dosages of tramadol and its active metabolite have not been conducted in this study. Including such measurements could have been beneficial, as evaluating the impact of genetic polymorphisms on the pharmacokinetics of tramadol in parallel with their effects on pain could provide a more comprehensive understanding of the drugs efficacy and safety profile. This approach would help correlate genetic variations with plasma levels of tramadol and its metabolites, thereby elucidating the mechanisms through which genetic polymorphisms influence pain relief. This must be discussed in the Discussion section.
â–¶ Response; The issue you pointed out had indeed been a concern from the planning stages of our research. However, no additional invasive tests beyond those mandated by the patients' normal treatment processes were permitted during the study execution, thereby limiting our measurement capabilities. We have noted this limitation in L-245.
# Comments 5; Line 170-3: The basal pain score showed a statistically significant difference between subject groups, with the details of OPRM1 ..between these genotype groups in this study (Table 3) : how the authors explain such results?
â–¶ Response; While further research is required to draw definitive conclusions, numerous studies have reported that individuals carrying the G allele exhibit a higher sensitivity to pain. Our study also observed a similar trend when measuring the basal pain scores, which is illustrated in Figure 4. However, in this study, no significant difference was observed in tramadol effect as assessed by changes in NRS (Numeric Rating Scale) based on the presence or absence of the G allele. As shown in references 39-52, the role of the G allele in analgesic effect varies across studies, making it difficult to reach a conclusive statement at this stage. As you suggested, a larger-scale study involving multiple institutions and more participants, which would include the measurement of plasma dosage of tramadol and its active metabolite, is expected to significantly contribute to elucidating a clearer relationship.
Thank you for your time and evaluation of our manuscript. We have done our best to reflect on your points thoroughly. We wish peace and wellness to your family.
Warm regards,
Min Woo Ha

Round 2
Reviewer 2 Report
Comments and Suggestions for Authors
Thanks for your replies to my previous comments, however, these are insufficient revisions of the manuscript, and the main points have not been properly addressed.
Overall the data doesn't seems to be of clinically relevance. PAtients took only one dose of tramadol. Authors suggests that this might be related to reluctance from patients. If this is true, than also the NRS outcome is questionable.
The authors should include in the manuscript most of the arguments used to reply to my previous comments and expand them.
This study may support a possible biological role of genetic variants in tramadol actions, but the use of these variants seems of little clinical value. This might be related to the biases in the population enrolled, however this point should be openly discussed in the manuscript.
Previous point on the definition of IM, NM UM phenotypes is not properly clarified, please expand the methods section to clarify how this were defined according to genotypes. and confirm wich is the expected ordinal or linear response to tramadol NM>IM>UM ? is that correct? Please clarify this in the methods.
Author Response
Thanks for your replies to my previous comments, however, these are insufficient revisions of the manuscript, and the main points have not been properly addressed.
#1. Overall the data doesn't seems to be of clinically relevance. Patients took only one dose of tramadol. Authors suggests that this might be related to reluctance from patients. If this is true, than also the NRS outcome is questionable.
â–¶# Answer; I appreciate your insightful observation. It has been noted that Muslims residing in desert regions often exhibited remarkable resilience. While they did not negate the experience of pain, it appeared that they considered it to be within tolerable limits. Given that the Numerical Rating Scale (NRS) for pain is inherently subjective, it is understandably challenging for researchers to ascertain the exact intensity of pain as reported by patients. This subjectivity is indeed a fundamental characteristic of such indices. Consequently, it would be unsubstantiated to dismiss the reliability of the NRS scores as self-reported by patients. Furthermore, the NRS scores observed in this study displayed trends that were consistent with those documented in existing literature. Specifically, patients possessing the G allele at rs1799971 (OPRM1 A118G) demonstrated significantly higher baseline NRS scores compared to their counterparts lacking the G allele. Additionally, the variations in NRS scores attributed to the effects of tramadol aligned with existing research, following the sequence of UM, NM, and IM.
Regrettably, the NRS score remains the most reliable measurement tool accessible to us at this time. However, should you have any methodologies for precisely measuring pain scores that circumvent patient bias, I would be most eager to incorporate them into future research endeavors. Furthermore, several studies have assessed the pharmacological effects by monitoring temporal changes in plasma concentrations of tramadol and its active metabolites through blood sampling. Due to our ethical guidelines, we refrain from any invasive procedures beyond the standard treatment regimen of the patients. This limitation is likely a point of contention raised by experts such as yourself. The specifics of this constraint are detailed in Line 247 of the text.
The authors should include in the manuscript most of the arguments used to reply to my previous comments and expand them.
This study may support a possible biological role of genetic variants in tramadol actions, but the use of these variants seems of little clinical value. This might be related to the biases in the population enrolled, however this point should be openly discussed in the manuscript.
â–¶# Answer; I have endeavored to address the concerns in my previous response, and I apologize if it did not fully meet your expectations. The issue of selection bias that you highlighted is indeed acknowledged and has been explicitly reiterated in Line 276 of the text.
Previous point on the definition of IM, NM UM phenotypes is not properly clarified, please expand the methods section to clarify how this were defined according to genotypes. and confirm wich is the expected ordinal or linear response to tramadol NM>IM>UM ? is that correct? Please clarify this in the methods.
â–¶# Answer; As you are undoubtedly aware, there exist multiple methodologies for distinguishing phenotypes based on the CYP2D6 genotype. In this study, we adopted the approach outlined by Gaedigk et al. in reference 19. In the paper detailing this method, the authors employed a modified Delphi method to achieve consensus on a standardized system for converting CYP2D6 genotypes to phenotypes, drawing from an international panel of CYP2D6 experts. Consequently, this method is widely utilized within this field of research. This information has been reiterated in Line 119 of the text.
Thank you for your thorough review and valuable feedback. We have carefully addressed all the comments and made the necessary revisions. We believe that these changes have significantly improved the clarity and robustness of our manuscript. We look forward to your positive response.
Warm regards,
Min Woo Ha

Reviewer 3 Report
Comments and Suggestions for Authors
The manuscript can be accept for publication
Author Response
We are delighted that you found our work worthy of publication and are grateful for your endorsement. Your constructive insights have undoubtedly contributed to enhancing the quality of our manuscript.
Thank you once again for your valuable time and effort in reviewing our work.
Sincerely,
Min Woo Ha
Round 3
Reviewer 2 Report
Comments and Suggestions for Authors
Dear Authors,
thanks for clarifying how you defined the phenotype-genotype (based on the previous literature). Now the methods are well-written.
The major problem you are not addressing is the limited clinical utility of these findings. Please take the following consensus as guidance (Nat Med. 2024 Jul;30(7):1874-1881. doi: 10.1038/s41591-024-03033-3) and try to describe in discussion the utility of the information from a patient's point of view. Since all the patients asked only for 1 dose, the genotype could not influence the clinical practice. Here you should be more open and discuss the limitation of your selected population, which for some reason did not ask for additional drugs to reduce the pain. Please add in discussion a paragraph on this in the limitation section and/or discuss the implication of this point.
Author Response
Thank you very much for sharing the latest insights published in July 2024. I have greatly benefited from reading it, and I believe it will also be valuable to our readers, which is why I have included it in the references. Additionally, I have addressed the limitations of this study in Section L-277, particularly regarding its insufficiency in contributing to precision medicine for clinical use of tramadol.
I have also noted the study's limitations concerning the interpretation of why all patients requested only a single dose of tramadol in Section L-251.
I wish peace and prosperity to your home.
